# Genome Plasticity of African Swine Fever Virus: Implications for Diagnostics and Live-Attenuated Vaccines

**DOI:** 10.3390/pathogens11020145

**Published:** 2022-01-24

**Authors:** Bonto Faburay

**Affiliations:** Scientific Liaison Services Section, Foreign Animal Disease Diagnostic Laboratory, National Veterinary Services Laboratories, Animal and Plant Health Inspection Service (APHIS), Plum Island Animal Disease Center, United States Department of Agriculture, New York, NY 11957, USA; bonto.faburay@usda.gov

**Keywords:** African swine fever virus, dsDNA virus, genetic mutation, diagnostics, live-attenuated vaccines

## Abstract

African swine fever (ASF) is a highly contagious transboundary viral hemorrhagic disease of domestic and wild pigs presenting a significant threat to the global swine industry. Following its introduction in Caucasus, Georgia, in 2007, the genome of the genotype II of African swine fever virus (ASFV) strain Georgia-07 and its derivatives accumulated significant mutations, resulting in the emergence of genetic variants within short epidemiological timescales as it spreads and infects different hosts in diverse ecosystems, causing outbreaks in Europe, South Asia, South East Asia and the Caribbean. This suggests that ASFV, with a comparatively large and complex DNA genome, is susceptible to genetic mutations including deletions and that although the virus is environmentally stable, it is genetically unstable. This has implications for the development of vaccines and diagnostic tests for disease detection and surveillance. Analysis of the ASFV genome revealed recombination hotspots, which in double-stranded DNA (dsDNA) viruses represent key drivers of genetic diversity. The ability of pox virus, a dsDNA virus with a genome complexity similar to ASFV, regaining virulence following the deletion of a virulence gene via gene amplification, coupled with the recent emergence and spread of live-attenuated ASFV vaccine strains causing disease and death in pigs in China, raise legitimate concerns around the use of live-attenuated ASFV vaccines in non-endemic regions to control the potential introduction. Further research into the risk of using live-attenuated ASFV in non-endemic regions is highly needed.

## Opinion

DNA viruses have been traditionally considered a group of viral pathogens whose genomes are stable and less susceptible to high spontaneous mutation rates. However, the widespread use of whole genome sequencing is increasingly revealing the accumulation of genetic variation in DNA genomes between population samples taken at different times. Although mutation rates and genome sizes tend to be inversely related, it has become evident that slow-evolving pathogens can accumulate variation throughout their larger genomes on a timescale comparable to that observed for RNA viruses [1]. Molecular evolutionary analyses revealed that large double-stranded DNA (dsDNA) viruses can indeed evolve quickly. An analysis of samples of African swine fever virus (ASFV) spanning 70 years revealed estimated evolution rates in the order of 10^−4^ substitutions per nucleotide per year, a value within the typical range exhibited by many RNA viruses [2]. This may provide a plausible explanation for the detection of multiple genetic variants of ASFV genotype II within short epidemiological timescales in outbreaks in China (within 2–3 years since 2018; [3]), South East Asia and Europe [4,5,6]. In fact, whole genomes of many bacterial and dsDNA genomes sequenced to date have been shown to accumulate novel mutations over timeframes of days to months, similar to those typically observed in RNA viruses [1]. Average intervals between transmission and nucleotide substitution events; i.e., when a novel mutation is observed, are determined to be months for dsDNA viruses, such as smallpox virus, days for HIV and Influenza A virus (H1N1) and a few years for *Mycobacterium bovis* [1,7,8,9]. This suggests that ASFV with a comparatively large and complex genome is susceptible to accumulating genetic mutations as it spreads and transmits within different host populations in different ecological environments. Together, and with the continuing identification of novel ASFV genotypes (24 p72 genotypes at present) in regions of endemicity in sub-Saharan Africa, where the virus is maintained through complex domestic and sylvatic transmission cycles provides an additional indication that ASFV, despite its ability to survive in different environmental matrices, such as feed [10], carcasses [11] and soil [12], is genetically unstable.

Although DNA viruses are generally reported to display low mutation rates largely due to their encoded high-fidelity DNA proofreading repair enzymes, the per-site evolutionary rate among complex dsDNA viruses can vary significantly along their genomes [13,14]. In fact, ASFV genomes uniquely encode the only known X-type polymerase (X Pol) and a DNA ligase with each exhibiting low fidelity [15,16]. The in vitro fidelity of the ASFV DNA ligase is, at present, the lowest known, whereas the genetic and phenotypic basis for the wide antigenic diversity exhibited by ASFV strains is yet to be determined [17,18,19]. However, the high level of antigenic diversity in field isolates is postulated to arise from the activity of the X Pol/DNA ligase, and some investigators have hypothesized that the X Pol-ligase system functions as a strategic DNA mutator in the base excision repair pathway, where rapid genetic drift occurs due to elevated DNA repair-based error [15,20,21]. With the lack of strong empirical knowledge about the fidelity of the ASFV DNA polymerase, in-depth studies into the mechanisms of DNA repair and DNA polymerase fidelity in ASFV are required in order to better understand ASFV genome plasticity and microevolution, which would enhance our capacity to develop efficacious vaccines and accurate diagnostic tests for disease detection and surveillance. Enzyme-linked immunosorbent assay (ELISA) is routinely used for disease surveillance and in conjunction with PCR for establishing population freedom from infection and facilitating international movement of animals. Commercially manufactured as well as Office International des Epizooties (OIE)-validated ELISAs using semi-purified p72 or a recombinant protein of p30, p54, pp62 and p72 have been developed in indirect and competitive formats [22]. Although, positive ELISA tests, especially in non-endemic countries, should be validated by a secondary confirmatory test, such as an immunofluorescent antibody test (IFAT) or immunoperoxidase test (IPT) [22,23]. Additionally, validated real-time PCR assays for detecting ASFV have been developed [24,25,26,27,28]. All these PCR assays target conserved regions of the viral p72 gene and appear to be capable of detecting all known ASFV genotypes. However, given the plasticity of the ASFV genome, molecular surveillance to monitor the conservation of diagnostic target sequences coupled with routine evaluation of the sensitivity of diagnostic tests in partnership with laboratories in endemic countries should be supported and carried out.

Next generation sequencing has provided the opportunity to perform a detailed analysis of whole genome sequences of ASFVs responsible for recent outbreaks around the world. The sensitivity of the analytical methods and pipelines could identify single nucleotide polymorphism (SNP) variants in outbreaks occurring within short epidemiological timescales. For example, since the first reported introduction of ASFV in Georgia in 2007, this genotype II variant has evolved over time into multiple genetic subvariants as it spread, causing outbreaks in Russia (2007), the Baltic States (2014), Poland (2014), Belgium (2018), China (2018), Vietnam (2019), Cambodia (2019), North Korea (2019), Mongolia (2019) and Germany (2020), and most recently in the Dominican Republic (2021) [29]. The parent Georgia 2007 isolate has since accumulated mutations, including potentially attenuating mutations, that could result in the emergence of less virulent variants (Wu et al., 2020) causing atypical or chronic infections that may potentially be missed by passive surveillance systems. It is hypothesized that these genetic mutations occur as an adaptive microevolution due to selection pressure as the virus replicates and infects different hosts in different agroecological environments. Indeed, intra-epidemic ASFV genome sequence variants have been characterized in outbreaks in Georgia [19] and China [3], providing additional insight into ASFV genome instability during natural infection. These observations provide a basis to postulate that the emergence or detection of multiple p72 genotype II variants within a short epidemiological timescale in recent outbreaks is the result of virus microevolution and (or by) adapting to new hosts and environments.

Molecular analyses of the ASFV genome revealed that the X64R, EP152R, EP153R, EP402R, EP364R and CP2475L genes are located in regions with very high genetic diversity [8], and it is known that gene regions that are associated with antigenic drift typically exhibit higher genetic diversity. A high level of variability is observed within the 35 kb at the 3′ end and the 15 kb at the 5′ end of the genome (170–190 kb) [30,31,32]. These two regions contain the multigene families (MGFs), which vary in number between isolates and give competence for virus variability by gene homologous recombination [19]. Variability is reported to also be generated by a change in the number of amino acid repeats, including the envelope protein p54 encoded by the E183L gene [33]. Indeed, a phylogenetic analysis suggests that recombination is a key driver of genetic diversity in dsDNA viruses, and ASFV genomes have shown more indel mutations than point base substitutions, which could be the result of ectopic homologous recombination, replication slippage, or retrotransposition [34]. These, coupled with the identification of recombination hotspots in the ASFV genome [8] and the characterization of recombinant ASFV variants among Italian and South African isolates [2], further add to the mounting evidence of the high genome plasticity of ASFV.

With the urgency to develop a protective ASF vaccine, several live-attenuated vaccine candidates have been developed [35,36] and are based on modifications to the genome of ASFV genotype II and, to a lesser extent, genotype I strains. Theoretically, ASF live-attenuated vaccines are engineered via the deletion of genes associated with viral virulence, host range and/or immunomodulation and include constructs with deletions of DP148R, thymidine kinase (TK), NL (DP71L), 9GL (B119L), 1177L, CD2v, UK and multiple members of multigene families 360 and 505 (MGF 360/505) [35,37,38,39,40,41,42,43,44,45]. However, concern regarding the potential risk for their use in non-endemic regions has been reported [46]. This concern is exacerbated by the emergence and spread of live-attenuated ASFV vaccine strains causing disease and death in pigs in China [47]—a development that provides further insight into possible genome instability of ASFV. Empirical evidence shows that double-stranded DNA viruses indeed exhibit genome instability. For example, next generation sequencing revealed herpes simplex virus laboratory samples exhibited new mutations after few transfers [48], further questioning the long-believed genetic stability of DNA viruses. The inverted terminal repeats of vaccinia virus genome are known to experience rapid changes in size, with diversity required in these regions for immune escape and for establishing infection in new hosts. This requirement is necessitated by variability in host-specific selective pressure exerted by host immunity [49]. Similarly, the plasticity of the poxvirus genome (a dsDNA genome with close similarity to the ASFV genome) was demonstrated for K3L and E3L virulence proteins [50,51]. When a vaccinia virus deleted for E3L in order to impose a strong selection pressure favoring gain-of-function mutations in the other protein kinase R suppressor, K3L [52], the virus became adapted to this deletion by increasing the copy number of the K3L gene, expanding its genome size by 10% in order to maintain its host virulence [52]. Low-frequency variants in the viral population carrying recombination breakpoints were suggested as the most likely triggers of these genomic expansion. Interestingly, the ASFV genome has been shown to harbor recombination hot spots [8] and inverted terminal repeats [53] and with a genome size, complexity and structural features similar to poxviruses, raises reasonable concern for vaccine safety [35] and potential reversion to virulence [46] through an unknown but similar mechanism of compensatory gene amplification for ASFV deletion mutants used as live attenuated vaccines in China and elsewhere in South Asia. Furthermore, the detection of ASFV genotype II variants harboring multiple natural mutations or deletions in recent disease outbreaks in China [54], the Dominican Republic and West Africa (Foreign Animal Disease Diagnostic Laboratory/APHIS, unpublished data) presents potential challenges for development of a protective vaccine against this highly devastating transboundary disease. While the likelihood for the occurrence of gene amplification may be speculative, research and the application of state-of-the-art high-resolution next generation sequencing and analysis algorithms in molecular epidemiological investigations provide unique opportunities to determine if such a phenomenon occurs.

## Data Availability

Not Applicable.

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
