# Peer review of "Genome Plasticity of African Swine Fever Virus: Implications for Diagnostics and Live-Attenuated Vaccines"

_pathogens, 2022, doi:10.3390/pathogens11020145_

Round 1

Reviewer 1 Report

This was an opinion of the author and was well written and included some interesting observations.

There were a couple of typos:

Lines 53 - 55 are currently in italics, change to normal font.

In line 80 the gene was written in italics (viral p72 gene) but none of the other gene names were italicized. Change to using one format only.

Author Response

Response to Reviewer 1

Reviewer: Lines 53 - 55 are currently in italics, change to normal font.

Response: Thanks for the comment. The words were italicized to put emphasis on the statement; however, in agreement with the reviewer the statement has be de-italicized in the revised manuscript. 

Reviewer: In line 80 the gene was written in italics (viral p72 gene) but none of the other gene names were italicized. Change to using one format only.

Response: The viral p72 gene has been changed to the normal format in the revised manuscript.

Reviewer 2 Report

Lines 8-9 - "...of domestic pigs" - this should be corrected. According to the Terrestrial Animal Health Code of the OIE - Suids are the only natural non-arthropod hosts for African swine fever virus (ASFV). These include all varieties of Sus scrofa (pig), both domestic and wild, and African wild suid species including warthogs (Phacochoerus spp.), bushpigs (Potamochoerus spp.) and the giant forest hog (Hylochoerus meinertzhageni).

Line 10 - maybe it needs to be mentioned there "the genotype II of African swine fever virus (ASFV)... "

Lines 71-73 - The sentence should be clarified. ELISA is used in the ASF endemic countries to monitor the disease status but for the disease surveillance only PCR testing should be used, as well for early detection of the disease, while due to ASF genotype II animals dies sooner than specific antibodies are developed. In the non-endemic by  ASF countries, ELISA testing is not useful and shall be not performed for early detection of ASF.

Lines 121-122 - it would be necessary to mention here on which genotype (genotype I or genotype II) live-attenuated vaccine candidates have been developed, while both genotypes differ in the clinical forms, pathogenicity and mortality/morbidity rates.

Lines 127-129 - live-attenuated ASFV vaccine strain used in China was developed of AFSV genotype I, however, since 2018 in China spreads ASFV genotype II.

From this opinion is not entirely clear, whether the author speaks on the ASFV of all genotypes in general, or in particular focuses on genotype I or genotype II. To include the clarification on the genotypes of ASFV will give more value to the opinion itself.

Author Response

Rebuttal to Reviewer 2

Reviewer: Lines 8-9 - "...of domestic pigs" - this should be corrected. According to the Terrestrial Animal Health Code of the OIE - Suids are the only natural non-arthropod hosts for African swine fever virus (ASFV). These include all varieties of Sus scrofa (pig), both domestic and wild, and African wild suid species including warthogs (Phacochoerus spp.), bushpigs (Potamochoerus spp.) and the giant forest hog (Hylochoerus meinertzhageni).

Response: I appreciate the comments made by the reviewer and reworded the statement in the revised manuscript to highlight both domestic and wild pigs as defined in the OIE Terrestrial Animal Health Code

Reviewer: Line 10 - maybe it needs to be mentioned there "the genotype II of African swine fever virus (ASFV)... "

Response: I agree with the reviewer’s suggestion and inserted “the genotype II of the African swine fever virus (ASFV) in the revised manuscript.

Reviewer: Lines 71-73 - The sentence should be clarified. ELISA is used in the ASF endemic countries to monitor the disease status but for the disease surveillance only PCR testing should be used, as well for early detection of the disease, while due to ASF genotype II animals dies sooner than specific antibodies are developed. In the non-endemic by ASF countries, ELISA testing is not useful and shall be not performed for early detection of ASF.

Response: I understand the reviewer’s concerns. Although PCR is a better tool for enhanced surveillance, both ELISA and PCR used in conjunction should provide a more robust disease surveillance strategy especially for the ability to detect infections caused by low virulent ASFV variants causing atypical clinical disease (without mortality) with seroconversion and may not be viremic to detect by PCR. To address the reviewer’s concern I have reworded the statement by including , ..and in conjunction with PCR”… (line 72 in the revised manuscript).

Reviewer: Lines 121-122 - it would be necessary to mention here on which genotype (genotype I or genotype II) live-attenuated vaccine candidates have been developed, while both genotypes differ in the clinical forms, pathogenicity and mortality/morbidity rates.

Response: I agree with the reviewer’s comments and included a statement to make a clarification that these live attenuated vaccines are based on genotype II and genotype I (lines 122 to 123 in the revised manuscript).

Reviewer: Lines 127-129 - live-attenuated ASFV vaccine strain used in China was developed of AFSV genotype I, however, since 2018 in China spreads ASFV genotype II.

Response: I understand the concern of the reviewer. However, several live attenuated ASF vaccines, including those based on genotype II backbones, have been developed and evaluated in the field tested in China (please see references below):

  • Chen W, Zhao D, He X, Liu R, Wang Z, Zhang X, et al. A seven-gene-deleted African swine fever virus is safe and effective as a live attenuated vaccine in pigs. Sci China Life Sci. 2020;63(5):623–34.
  • Teklue T, Wang T, Luo Y, Hu R, Sun Y, Qiu HJ. Generation and evaluation of an African swine fever virus mutant with deletion of the CD2v and UK genes. Vaccines. 2020;8(4):763.

There is no information to indicate that these vaccines or others based on genotype II have not been used by farmers/pig producers and contributes to the spread of genotype II strains in China.

Reviewer: From this opinion is not entirely clear, whether the author speaks on the ASFV of all genotypes in general, or in particular focuses on genotype I or genotype II. To include the clarification on the genotypes of ASFV will give more value to the opinion itself.

Response: I understand the reviewer’s concern. It is correct I made mentioned genotype II but that is for making a point as a reference. However, the opinion is about ASFV generally. This is highlighted in my statements …..”Together, and with the continuing identification of novel ASFV genotypes (24 p72 genotypes at present) in regions of endemicity in sub-Saharan Africa, where the virus is maintained through complex domestic and sylvatic transmission cycles provides additional indication that ASFV despite its ability to survive in different environmental matrices such as feed , carcasses  and soil, is genetically unstable”.  (lines 50 to 55).

Furthermore, I have also inserted a statement mentioning that the modified live attenuated vaccines are based genomes of both genotype II and genotype I (lines 122 to 123). This should help clarify that the opinion is about ASFV genome generally and includes h genotype I and genotype II.

Reviewer 3 Report

This is a very controversial paper without any experimental proof. Recently the GARA (Global African Swine Fever Research Alliance) organized the webinar Troubleshooting Techniques for Full Genome Sequencing of African Swine Fever and one of the topics was addressed to effective use of very limited resources for full genome sequencing, stop duplicating activities with Georgia 2007 genetic line and pay more attention to other ASF genotypes to accelerate the vaccine development.

Concerning the genetic changes from the point of view of diagnostics: all gap analysis shows that diagnostic technics fit to purpose and there are no significant gaps in ASF diagnostics.

Changes in variable regions of the ASFV genome are obvious and this is the reason why these parts of the genome are named "variable". The situation with attenuated strains appearance in China is not clear at this moment but deletions in six different parts of ASFV genome can not be explained by natural selective pressure.

In fact after almost 15 years "Georgia 2007" strain circulation in the Eurasia virus, in general, did not change and the statement that ASFV is "genetically unstable" from a practical point of view has no evidence.

Also important to keep in mind that the original hows of ASF - ticks but not pigs, and maybe these genome changes affect virus-host interaction in the case of ticks, but forSus scrofa till now it does not play a significant epidemiological role.

Author Response

Rebuttal to Reviewer 3

This is a very controversial paper without any experimental proof. Recently the GARA (Global African Swine Fever Research Alliance) organized the webinar Troubleshooting Techniques for Full Genome Sequencing of African Swine Fever and one of the topics was addressed to effective use of very limited resources for full genome sequencing, stop duplicating activities with Georgia 2007 genetic line and pay more attention to other ASF genotypes to accelerate the vaccine development.

Reviewer: Concerning the genetic changes from the point of view of diagnostics: all gap analysis shows that diagnostic technics fit to purpose and there are no significant gaps in ASF diagnostics.

Response: I agree with the comments of reviewer; and indeed, the fact current diagnostic tests especially molecular seem to perform satisfactorily (i.e. the existence of major significant gap) has been addressed in the revised manuscript (lines 71 – 80). Herein, it is mentioned “all these PCR assays target conserved regions of viral p72 gene and appear to be capable of detecting all known ASFV genotypes. However, there are ELISAs based on p54 protein, and variability is reported to be also generated by a change in the number of amino acid repeats, this envelope protein by the E183L gene [see reference 33] (lines 111-113 in the revised manuscript). This evidently could impact serological tests; although serological tests based on other targets may less affected. Thus, what the Opinion is highlighting is that genetic mutation is occurring in the ASFV genome, its potential implications on diagnostic performance in the future and the need for vigilance/genomic surveillance.

Reviewer: Changes in variable regions of the ASFV genome are obvious and this is the reason why these parts of the genome are named "variable". The situation with attenuated strains appearance in China is not clear at this moment but deletions in six different parts of ASFV genome cannot be explained by natural selective pressure.

In fact after almost 15 years "Georgia 2007" strain circulation in the Eurasia virus, in general, did not change and the statement that ASFV is "genetically unstable" from a practical point of view has no evidence.

Response: I agree with the reviewer about the variability in the ASFV genome, and the fact that it is not clear what is driving the genetic changes in the ASFV genome. However, it is abundantly clear that the ASFV genome undergoes genetic microevolution (we/FADDL/USDA have seen that in the more 50 genomes sequenced in the Dominican Republic Outbreak and in a local outbreak in West Africa/Nigeria) among genotype variants within short epidemiological timescales; and also the that Georgia 2007 has undergone genetic variation which has been detected as early as 2008 (following introduction in 2007) (please see reference 19) and with subsequent emergence of genetic genotype II variants with significant genetic mutations including deletions as the virus spreads from Europe to Asia (these are derivatives of Georgia2007). Our lab has demonstrated this via SNP analysis (in preparation for publication).  

In fact, Chinese scientists have confirmed the emergence of genotype II variants (derivatives of Georgia 2007) with mutations and this could be an ongoing phenomenon. This is highlighted in the following statement from an article in Reference 52:   

The scientists did a surveillance in 7 Chinese provinces from June to December 2020. They took 3,660 samples from farms, slaughterhouses and disposal plants in Hebei, Heilongjiang, Hubei, Inner Mongolia, Jilin, Liaoning and Shanxi provinces. They isolated and characterized 22 viruses belonging to the genotype II ASFv. All 22 had mutations, deletions or replacements in comparison to “HLJ/18”, the earliest isolate in China”.

Reviewer: Also, important to keep in mind that the original hows of ASF - ticks but not pigs, and maybe these genome changes affect virus-host interaction in the case of ticks, but forSus scrofa till now it does not play a significant epidemiological role.

Response: I thank the reviewer for this comment and insight. However, the comment made above is not a subject of my current analysis in this Opinion paper; nonetheless it is an important subject that should be examined in the future.    
